

# Continental drivers of ammonium and nitrate in Australian soil under different land uses

Juhwan Lee[1], Gina M. Garland[2], and Raphael A. Viscarra Rossel[1]

[1]CSIRO Land and Water, GPO Box 1700, Canberra ACT 2601, Australia
[2]Agroscope, Zurich 8046, Switzerland

**Correspondence:** Juhwan Lee (juhwan.lee@csiro.au)

**Abstract.** Soil N is an essential element for plant growth, but its mineral forms are subject to loss to the environment by leaching and gaseous emissions. Despite its importance for the soil-plant system, factors controlling soil mineral N concentrations over large spatial scales are not well understood. We used $NH_4^+$ and $NO_3^-$ concentrations (0–30 cm depth) from 469 sites across Australia, and determined soil controls on their regional variation. Soil mineral N varied regionally but depended on the

different land uses. In the agricultural region of Australia, $NH_4^+$ tended to be depleted ($4.9 \pm 4.8$ vs. $5.6 \pm 9.0$ mg N kg$^{-1}$) and $NO_3^-$ was significantly enriched ($6.0 \pm 9.2$ vs. $3.8 \pm 9.9$ mg N kg$^{-1}$), compared to the non-agricultural ecological region. The relative importance of soil controls on mineral N in the agricultural region, identified by the model trees algorithm Cubist, showed that $NH_4^+$ was affected by total N, cation exchange capacity (CEC) and pH. In the ecological region, $NH_4^+$ was affected by CEC and pH, but also organic C and total P. In each of the regions, $NO_3^-$ was primarily affected by CEC, with more complex

biophysical controls. In both regions, correlations between mineral N and soil C:N:P stoichiometry suggest that more $NH_4^+$ was found in P-depleted soil relative to total C and total N. However, our results showed that only in the other ecological region, $NO_3^-$ was sensitive to the state of C and its interaction with N and P. The models helped to explain 36–68% of regional variation in mineral N. Although soil controls on high N concentrations was highly uncertain, we found that region-specific interactions of soil properties control mineral N concentrations and therefore it is essential to understand how they alter soil

mechanisms and N cycling at large scales.

## 1 Introduction

Nitrogen is an essential element and one of the main factors limiting the functioning and productivity of various soil-plant systems. This is particularly the case in cropping-dominant systems, where annual crop productivity often depends on rotations with $N_2$-fixing legumes or continuous addition of chemical fertilizer or organic matter. External N enters the soil (e.g.,

atmospheric N deposition and fertilizer N) and plants (e.g., $N_2$ fixation), and cycles through them in various chemical forms or states, mostly mediated by soil microbes. Mineralization (ammonification), microbial immobilization, nitrification and denitrification determine the availability of $NH_4^+$ or $NO_3^-$ that can be taken up by plants or lost to the environment, which in turn contributes to the regulation of terrestrial C sequestration (Post et al., 1985; Rastetter et al., 1997; Luo et al., 2004; Thornton et al., 2007; Ciais et al., 2013). Biotic and abiotic transformations of N from mineralized soil organic matter or derived from



external soil inputs are spatially and temporally variable. In addition, different mineral forms of N often show different concentration patterns and relative importance in terms of contribution to total N stocks across a broad range of land uses and management practices due to different climatic and ecological conditions (Johnston, 1993; Farley and Fitter, 1999).

Managed soil, especially that used for agricultural production and grazing, is often characterized by N imbalances and re-
active N losses to the environment (Vitousek et al., 1997; Ciais et al., 2013; Fowler et al., 2013). One of the consequences of continuous cropping on the N cycle is the increased emissions of ammonia, N oxides ($NO_x$), and nitrous oxide ($N_2O$). Nitrous oxide is a potent greenhouse gas and ozone-depleting substance, while $NO_x$ produces ozone in the troposphere (Ravishankara et al., 2009). Other concerns include leaching of $NO_3^-$ to groundwater, which may contribute to terrestrial and coastal eutrophication (Conley et al., 2009). Accordingly, there are increasing demands to measure or simulate the fate of residual mineral N
across agricultural areas as well as in the soil under native vegetation, which can be conceptually linked to the loss of N to air and water upon land-use conversion. Indeed, there is a strong need for large-scale soil information from which to accurately define current baselines and to determine the state of soil mineral N and its response to soil conditions.

Soil mineral N concentrations can be highly variable and uncertain because multiple reaction pathways and oxidation states of N processes exist (Butterbach-Bahl et al., 2013). Nitrogen transformation processes in soil are controlled by the complex
interactions of soil variables, such as substrate availability, pH, temperature and water content, which are affected by ecological factors (i.e., soil, climate, vegetation and topography) and past management history. In particular, soil properties related to the storage and quality of soil organic matter and its management could affect the levels and variation in soil $NH_4^+$ and $NO_3^-$ (Vagstad et al., 1997). Soil N transformations also involve the biogeochemistry of other nutrients, such as C and P, which are key constituents of organic molecules. In particular, regular N input into the soil, including plant N returning to soil as residues
or decomposing roots, is important for the maintenance of soil organic C (SOC), and therefore the availability of soil N to plants, and closely interacts with the terrestrial organic C sink (Rastetter et al., 1997; Zaehle, 2013). Stiles et al. (2017) showed that N fertilizer addition alone resulted in an increase in both SOC and total N, but the combined addition of N and P had the opposite effect. Overall, N cycling is closely linked to soil C sequestration and stabilization, but C-N cycle coupling can be complicated through changes in soil P. Glendining et al. (2011) found that, globally, SOC and the C:N ratio of soil organic
matter are important controls of total N in soil, although large uncertainties remain. Therefore, the effects of soil N and P and their elemental interactions on mineral N dynamics are important (Booth et al., 2005; Taylor and Townsend, 2010), but they have not been widely explored across large spatial scales.

Differences in inherent soil properties, as well as land use, cause a shift in dominant forms of soil N and its distribution across sites. Land uses are generally characterized by vegetation types, dominant sources of soil N input and the levels of soil
disturbance and often coincide with climate and ecological conditions. Therefore, it may be possible to specifically compare the patterns of mineral N across broad regions by land use. The amount of N stored in soil in organic and mineral forms is considered a limiting factor for the primary productivity of many ecosystems. How primary productivity is managed affects the rate at which soil N and other essential nutrients are replenished. Yet, there is often a long-term tendency of ecosystem functions, such as biodiversity, to decline with external additions of reactive N through atmospheric deposition and fertilizer




inputs (De Schrijver et al., 2011). It is therefore important to determine soil mineral N patterns and their controlling factors, but there are limited soil N data available for understanding plant and soil processes at different scales.

Since primary soil controls on mineral N are largely unknown at the continental-scale of Australia, it is relevant to measure and understand how the state of soil N differs between regions, where diverse dominant land-use conditions exist. Our aim

here is to determine soil controls on the continental variation in soil $NH_4^+$ and $NO_3^-$ concentrations between and within the agricultural and non-agricultural ecological regions of Australia. We focused on a set of soil attributes and assessed their relative importance for determining the large-scale drivers of mineral N concentrations by land use.

## 2   Materials and Methods

### 2.1   Data sets

We used two soil data sets for our analysis. First, the Biomes of Australian Soil Environments (BASE) project (Bissett et al., 2016) provides soil and other contextual data across the continent, originally developed for assessment of soil biodiversity (available at https://data.bioplatforms.com). The BASE project performed soil sampling and analysis, as described in Bissett et al. (2016). For this study, we obtained the concentrations of soil $NH_4^+$, $NO_3^-$, total organic C (TOC), exchangeable cations ($Al^{3+}$, $Ca^{2+}$, $Mg^{2+}$, $K^+$, and $Na^+$), texture and pH (in $CaCl_2$) from BASE, which were measured at the 0–0.1 m and 0.2–0.3 m

depths from over 650 sites across Australia between 2011 and 2016 (Fig. 1). Each sample was collected from a site (25 m × 25 m) that represents a unique environmental condition. Site soil sampling was done in a destructive manner, and measured at a single time point because no further samples were resourced. Spatial distribution of $NH_4^+$ and $NO_3^-$ concentration can be different and temporally unstable for each site (Giebel et al., 2006). So far, this is the most comprehensive data describing soil mineral N pools across different soil and land-use types of Australia at the continental scale, including a wide range of

agricultural soil conditions. At the scale of study, we assumed that spatial patterns of soil mineral N remained persistent with time. This assumption was based on the number of sites across which the effect of confounding factors may be minimized. However, this source of uncertainty needs to be tested by sampling spatial patterns at different times of the year. The size of sampling area was fixed to match the smallest grid size of legacy soil attribute maps in Australia. Although fine-resolution (< 25 m) soil heterogeneity was not addressed, the BASE data of soil $NH_4^+$ and $NO_3^-$ concentrations could be extended with key

soil attributes previously produced from national mapping efforts but not measured in BASE. The exchangeable cations were summed to estimate cation exchange capacity (CEC).

Second, to complement the BASE data set, we used Australian soil attribute data extracted from recently produced soil maps with robust uncertainty estimates at a pixel resolution of 90 × 90 m (or 3-arcsecond) (Viscarra Rossel et al., 2015). The maps that we used were the concentrations of total N (TN) and total P (TP), bulk density (BD), and available water capacity (AWC).

Each of the Australian soil attribute maps were independently produced by 3D spatial modeling that combined historical soil data and estimates derived from visual and infrared soil spectra. The approaches used to produce the maps were described in detail in Viscarra Rossel et al. (2015). When the soil map data were produced at the fine spatial scale, auxiliary environmental data were already considered in the spatial modeling. These variables represent proxies for the main environmental factors of





soil formation, which were related to parent material, climate, biota and vegetation, and terrain and landscape position. For this study, a weighted average over the 0–5, 5–15, and 15–30 cm depths was calculated for each of the mentioned soil variables. The depth-average values of each soil attribute were extracted using the coordinates from the 469 sites at which both $NH_4^+$ and $NO_3^-$ concentrations were measured. Extracting grid values to each geographic location and compiling data was done in the

Geographic Resources Analysis Support System (GRASS) GIS 7.2 (http://grass.osgeo.org).

## 2.2 Data analyses

The data set was screened for any missing values of land uses and soil properties. A total of 469 sites were retained for data analyses (Fig. 1), and we performed the analyses on (1) all samples, (2) the samples from sites used mainly for agricultural production and grazing (hereafter referred to as the "agricultural" region), and (3) the samples that were outside of the produc-

tion zones (referred to as the "ecological" region) (Fig 1). The agricultural region (160 sites) covers the main grain-cropping zones of Australia, which differs by climate and soil regimes, and farming practices. If the concentration of soil mineral N in the data was reported as the value at or below the detection limit of 1 ppm, it was replaced with 0.5 mg N kg$^{-1}$ (a median of the detection limit). The concentration of soil mineral N was averaged over the depths and then log-transformed to approximate normality. Mean comparisons of the log-transformed data were performed between the regions at the 0.05 significance level (p

< 0.05) using analysis of variance (ANOVA). Tukey's HSD test was used to compare the measured mineral N concentrations among the multiple land uses. The data set covered four general land uses: 1) conservation and natural environments, 2) production from dryland agriculture and plantations, 3) production from irrigated agriculture and plantations and 4) production from relatively natural environments. The general land use classes included 28 detailed land uses, according to the Australian Land Use and Management Classification (Supplementary Table 1).

All soil properties, except for pH, were log-transformed for further modeling. For each of soil $NH_4^+$ and $NO_3^-$ measurements, data analyses were performed on the whole data, including 22 samples for $NH_4^+$ and 170 samples for $NO_3^-$ that were measured under the detection limit. Only 3 and 42 of these samples for $NH_4^+$ and $NO_3^-$ were located within the agricultural region, respectively. Empirical models of soil mineral N as a function of the selected soil properties were built using the machine learning algorithm Cubist (Quinlan, 1992). Cubist is a form of rule-based decision tree with piecewise linear models. Models

were developed and evaluated by 10-fold 50-repeated cross-validation. Model performance was assessed using the coefficient of determination ($R^2$), the root mean squared error (RMSE), the mean error (ME), and the standard deviation of the error (SDE). In Cubist, the number of committees was fixed as one to avoid producing complex models, but the number of nearest neighbors was optimized using the RMSE of resampling results. The optimized models consisted of 1–8 different rule sets. The relative importance of each soil variable was assessed based on the usage of each individual variable in the rule conditions

and the model for Cubist. The cut-off was set at 80% as the probability to be essentially used in either the rule conditions or the linear model. In addition, the sensitivity of important soil variables was tested with 5, 10, and 20 committee models. The same data analysis and regression approach were applied on the data sets for the cropping and ecological regions separately, and then a Pearson correlation analysis was performed, accordingly. All statistical analyses and Cubist modeling were performed in R





version 3.4.3 (R Core Team 2017). Functions from the "Cubist" (version 0.0.21) and "caret" (version 6.0.76) packages were used.

## 3   Results

### 3.1   Soil mineral N in relation to large-scale land use

The concentrations of $NH_4^+$ and $NO_3^-$ in the soil were $5.3 \pm 7.8$ and $4.5 \pm 9.7$ mg N kg$^{-1}$ (mean $\pm$ SD), respectively, across all sites (Fig. 2). The median concentrations of $NH_4^+$ and $NO_3^-$ were 3.5 and 1.5 mg N kg$^{-1}$. The corresponding coefficient of variation was 147% for $NH_4^+$ and 215% for $NO_3^-$ across the sites, showing wide variation in the measured soil mineral N. The mean sum of $NH_4^+$ and $NO_3^-$ was $9.9 \pm 12.5$ mg N kg$^{-1}$, with the maximum up to 123 mg N kg$^{-1}$. Compared to the ecological region, the soil in the agricultural region had similar concentrations of $NH_4^+$ ($4.9 \pm 4.8$ vs. $5.6 \pm 9.0$ mg N kg$^{-1}$),

but significantly ($p < 0.05$) larger concentrations of $NO_3^-$ ($6.0 \pm 9.2$ vs. $3.8 \pm 9.9$ mg N kg$^{-1}$). There were a number of sites with relatively small concentrations of mineral N, particularly $NO_3^-$. If we only considered soil $NO_3^-$ concentrations above the detection limit, the mean and median $NO_3^-$ concentrations then increased to 8.0 and 4.5 mg N kg$^{-1}$ in the agricultural region and to 6.1 and 2.5 mg N kg$^{-1}$ in the ecological region, respectively. Nevertheless, the regional patterns of $NH_4^+$ and $NO_3^-$ remained consistent. The sum of soil mineral N showed a regional difference similar to that of $NO_3^-$. For all sites, the ratio of

$NH_4^+$-N to $NO_3^-$-N was $4.8 \pm 7.7$ or the median of 2.0. Specifically, soil $NH_4^+$ was identified as a dominant mineral form of N at 309 sites. The $NH_4^+$ and $NO_3^-$ fractions of TN were $0.5 \pm 0.5\%$ and $0.5 \pm 1.0\%$, respectively, and comprised together a mean range of 1–2% of TN in the soil at the sites. The $NO_3^-$ fraction of TN ranged between 0.01% and 10.0% and appeared to be more variable than that of $NH_4^+$, which had the range of 0.05–5.8%. The mean $NH_4^+$-N : $NO_3^-$-N ratio was significantly lower in the agricultural region (mean 3.4) than the ecological region (5.5). This also corresponded to a significant regional

difference in the $NO_3^-$ fraction of TN.

The differences in $NH_4^+$ and $NO_3^-$ concentrations between the regions were attributed to the differences among both broad and detailed land uses (Table 1). Soil $NH_4^+$ concentrations showed no difference among the broad land uses, but there was more variation for conservation and natural environments compared to the other land uses. Relatively low $NH_4^+$ concentrations were found in the soil used for production from relatively natural environments. Overall, also considering the limited data, little or

no differences were found between the detailed land uses. Large soil $NO_3^-$ concentrations generally resulted from agricultural production, compared to the soil in conservation and natural environments or used mainly for production from relatively natural environments. This pattern was generally found for detailed land uses as well.

### 3.2   Soil controls on the concentrations of mineral N

The relative importance of each of the soil variables as the primary controlling factor over soil mineral N are shown in Fig. 3.

Across all sites, the soil properties appeared to have effects on $NH_4^+$ concentrations, except the sand and silt fractions, BD, and AWC. Especially, the variation in soil $NH_4^+$ concentrations was consistently related to pH (Supplementary Fig. 1). The





concentrations of $NO_3^-$ were affected by primarily affected by CEC, and to a lesser extent by soil properties, similar to the controls on soil $NH_4^+$. In addition, BD was identified as a potential driver for $NO_3^-$ only. In the agricultural region, soil $NH_4^+$ was controlled by TN, CEC, and pH, while TN and CEC as well as TOC, sand fraction, and AWC were important controlling factors of $NO_3^-$. Among these soil properties, only pH and CEC showed consistent large-scale effects on mineral N (Supplementary

Fig. 1 and 2). There was no effect of TOC and TP on $NH_4^+$, but their effects were important in more complex models. In the ecological region, soil $NH_4^+$ was affected by TOC, TP, CEC, and pH. Soil $NO_3^-$ was affected by all selected soil variables in the same region, where BD, CEC and TOC were the most important factors. There was some effect of TN, but TP and pH had relatively less important contribution in contrast to its relative importance for the soil controls on $NH_4^+$. In general, based on Pearson's correlation coefficients, the concentrations of $NH_4^+$ and $NO_3^-$ were significantly correlated with the soil variables

identified by Cubist (Table 2). The exception was CEC, which was not correlated with but selected as important for $NH_4^+$ in the agricultural region. A similar case was found between $NO_3^-$ and TN or BD when accounting for all sites or regions.

After cross-validation, Cubist models were able to explain $60 \pm 11\%$ of the measured variation for $NH_4^+$ and $42 \pm 13\%$ for $NO_3^-$ in the soil across all sites (Table 3). The models were evaluated by considering all sites together, as well as the sites in each of the regions separately. Specifically, the RMSE and SDE of the region-specific models tended to decrease, showing

overall improvement in accuracy and precision. The Cubist models appeared to reasonably cover a range of measured $NH_4^+$ and $NO_3^-$ concentrations for each selected region (Fig. 4). However, the model failed to reproduce a high range of measured mineral N values when considering all sites or each of the regions. Therefore, the soil factors identified by the model may have unstable effects on these high values.

### 3.3   Relationships between mineral N and soil C, N, and P stoichiometry

Depending on the specific region, soil nutrients had distinct effects on the level of each mineral form of N. In the agricultural region (Fig. 5), soil $NH_4^+$ was directly related to TN and TOC to a lesser extent TP, with significant effects from the interaction of total soil nutrients ($p < 0.05$). Soil $NO_3^-$ was significantly related to TP only in the agricultural soil ($p < 0.05$). However, no significant relationships between $NO_3^-$ and all soil elemental ratios, and the model, suggest that the concentrations of $NO_3^-$ were insensitive to changes in elemental TP but indirectly related to TOC or TN. In the ecological region (Fig. 6), each of

the total nutrient concentrations was significantly related to the distribution of soil $NH_4^+$ in a similar manner to the soil in the agricultural region. This suggests that TOC was a main controlling factor, but the effects of elemental interactions would be potentially important, also corresponding to those from the modeling. Specifically, the levels of $NH_4^+$ increased in the P-depleted soil relative to the other nutrients in both regions. Therefore, there may be an essential but indirect effect of TP on soil $NH_4^+$, which was mostly masked by the relative composition of TOC and TN in the soil. For soil $NO_3^-$ in the ecological

region, the effect of TOC may have been through the interaction of TN and TP, even though the concentrations of $NO_3^-$ were not correlated with the TN : TP ratio.



## 4 Discussion

### 4.1 Continental variation in soil $NH_4^+$ and $NO_3^-$

The distribution of $NH_4^+$ and $NO_3^-$ in the soil, and the ratio of $NH_4^+$-N to $NO_3^-$-N show that $NH_4^+$ is a predominant source of N for plant uptake or other biological processes across the sites, compared to $NO_3^-$. This suggests that the risk of loss from

leaching and denitrification may not be large or evident at the scale of the study or from the sparse data set. The sum of $NH_4^+$ and $NO_3^-$ at the sites might be useful to approximately set the potential limits of inherent soil N availability, and thus the relevant limits for mineral N management particularly in the agricultural region of Australia. However, it was again based on limited soil data across a vast area of the continent and should be interpreted with caution. In our case, the distribution of $NO_3^-$ was characterized by relatively low values or values under the detection limit, which may suggest that those soils were

depleted in $NO_3^-$. On the other hand, large-scale variation in soil $NO_3^-$ and total mineral N concentrations should be considered when making implications for regional effects of land use patterns on soil N dynamics. The $NO_3^-$ fraction, relative to $NH_4^+$, of TN substantially increased if the soil with small $NO_3^-$ concentrations was excluded (data not shown). There may be also extreme values for $NH_4^+$ and $NO_3^-$, usually higher than 45–50 mg N kg$^{-1}$ approximately equivalent to 150–200 kg N ha$^{-1}$, representing potential hot spots. Such a wide range of $NH_4^+$ and $NO_3^-$ concentrations, on top of spatially scarce soil data,

presents a challenge to determine the responses of varying mineral N to soil factors under different land-use conditions.

We found that different land uses between the regions led to complex, but consistent regional patterns of soil $NH_4^+$ and $NO_3^-$. The soil in the agricultural region was characterized by relatively smaller $NH_4^+$ but significantly larger $NO_3^-$ concentrations than the soil in the ecological region, generally from conservation areas and natural environments (Fig. 2). However, each site in the agricultural region is not necessarily dominated by agricultural soil but represents more modified landscapes. This

region accounts for most of the soil conditions affected by land uses particularly related to agricultural production and grazing. Overall, more $NO_3^-$ appeared to be accumulated in the soil under agricultural conditions because of the significant mean difference in the sum of mineral N between the regions. The relative accumulation of soil $NO_3^-$ was also supported by the ratio of $NH_4^+$ to $NO_3^-$, which was considerably low in the soil from the agricultural region compared to the ecological region. The soil receiving high N inputs from external sources and recycled N may have more $NO_3^-$ in the balance of soil $NH_4^+$ and $NO_3^-$

pools (Watson and Mills, 1998). In the agricultural region, historical soil N input is known to enhance potential N nitrification over time, although it is further complicated by continuous soil disturbance (e.g., intensive tillage) at various levels (Angus and Grace, 2017). In addition, the soil's capacity to supply $NH_4^+$ through N mineralization may have been decreased with soil organic matter decline in response to continuous soil disturbance (Viscarra Rossel et al., 2014). Regional balance of soil mineral N would in part depend on preferential N uptake in main crops (Haynes and Goh, 1978; Gastal and Lemaire, 2002;

Andrews et al., 2013). However, this effect has not been shown for many crops grown in the region, and thus more data are needed to further confirm this.

Land-use types play a role as the major sources of regional differences in soil $NH_4^+$ and $NO_3^-$ across the sites (Table 1). It is therefore promising to split the continental data on mineral N into the region-specific variation in N based on land uses, if the current land uses were maintained with minimal spatial and temporal changes. Land uses and associated conditions





can geographically constrain or be constrained by each other, contributing to regional differences in soil characteristics and N transformation rates (Booth et al., 2005). As shown above, the amount of $NH_4^+$ or $NO_3^-$ stored in the soil was distinctly characterized between the regions, subject to different dominant land uses. The land-use types affected by production activities have been driven by N inputs and eventually led to enrichment of soil $NO_3^-$, also as suggested by the differences in mineral

N between the agricultural and other ecological regions. $NH_4^+$ was depleted in the soil within low-input or relatively natural environments used for production compared to the soil under conservation and natural environments not used for agricultural purposes. This has important implications because resource-based production systems with little or no input (e.g., grazing of native vegetation) may not be sustainable compared to input-driven production systems. The effect of agricultural land uses on the levels of $NH_4^+$ and $NO_3^-$ may depend on the interactive effect of soil management and external N input from anthropogenic

sources (Fueki et al., 2010). Similarly, soil $NH_4^+$ and $NO_3^-$ concentrations under cropping were about 3.4 and 6.2 mg N kg$^{-1}$, respectively, showing the decline in $NH_4^+$ but enrichment of $NO_3^-$. The levels of soil mineral N were highly variable in the protected or natural environments, particularly for $NH_4^+$, similar to the patterns between the regions. Presumably, this was due to the differences in the amount and quality of biomass input under natural vegetation and crop production, leading to different soil organic matter decomposition (source) versus N immobilization (sink) (Post et al., 1985; Post and Kwon, 2000).

Among ecological factors, the climatic variables, such as precipitation and temperatures, are known to strongly limit soil N storage across spatial scales (Post et al., 1985; Liu et al., 2017). In this study, however, we did not determine how soil mineral N status would change under other climatic and ecological conditions in Australia. This was because most of the sites were located in arid or temperate ecological zones. Given the number of the sites, the extent and support of measured soil data did not fully represent a possible range of soil conditions. Therefore, further measurements are needed to account for diverse

agro-ecological settings.

## 4.2  Soil controls over the measured variation in $NH_4^+$ and $NO_3^-$

Interactions between soil variables and related processes operating on multiple aspects of the N cycle are often difficult to understand at the continental scale. Much less is known about the large-scale effect of land-use patterns on soil mineral N dynamics. In this study, we used the measured data as well as the high-resolution soil data from model predictions that are

reliable and accurate with small to moderate uncertainty (Viscarra Rossel et al., 2015; Viscarra Rossel and Bui, 2016). Although potential uncertainties exist in the data as a proxy for "real" soil conditions, multiple soil variables were significantly related to $NH_4^+$ and $NO_3^-$ concentrations in the soil at all sites and between the regions (Table 2). In Cubist models, a comparable set of the soil variables acted as the controlling drivers on the distribution of $NH_4^+$ and $NO_3^-$ across the sites (Table 2 and Fig. 3). Rule-based models are practical to interpret and extrapolate plant N availability and N losses under various conditions. These

common soil drivers suggest that similar mechanisms drive $NH_4^+$ and $NO_3^-$ to some extent, sequentially linking them together. For example, CEC contributed to the potential to retain both $NH_4^+$ and $NO_3^-$. In addition, there were probably more complex interactions of soil controls on $NO_3^-$ concentrations, for example, including additional effects of soil texture. Since the selected soil variables can potentially determine soil mineral N availability and losses, the accurate measurement and mapping of soil attributes are important for continental simulations of soil N dynamics. Some of the soil variables, particularly CEC, may have




indirect effects on soil mineral N, presumably confounded by other predominant drivers, such as soil texture and BD (Saxton et al., 1986; Dunne and Willmott, 1996).

It is important to note that the relative importance of soil controlling factors was different between the agricultural and ecological regions. These variables (e.g., TP) previously identified as important across the sites could serve as conditional controls between the regions, depending on dominant land-use and vegetation types. Particularly for the soil in the agricultural region, the concentrations of soil $NH_4^+$ and $NO_3^-$ tended to be controlled by only 3–5 primary drivers, with the interactions of less important variables. These included pH for $NH_4^+$ and CEC for $NO_3^-$ in the soil. Total organic C, TN, and TP had varying degrees of importance that were specific for each mineral form of N between regions. In the agricultural region, $NH_4^+$ depended on TN and thus the distribution and sources of organic matter and potential N availability in the soil. Instead, $NH_4^+$ retention capacity may become important through pH and CEC effects. In the ecological region, soil $NH_4^+$ might depend on mineralization of organic matter sources, but less limited by TN or TP under near steady-state conditions. Soil $NO_3^-$ was affected by TOC and TN in the agricultural region, where it could be more affected by the capacity of the soil to immobilize residual $NO_3^-$ if ample N supply was assumed. In contrast, $NO_3^-$ was affected to a certain extent by TOC and TN and the effect of TP remained relatively higher in the ecological region, receiving limited soil inputs. Thus, soil $NO_3^-$ may presumably be associated with organic matter and N mineralization rates. At this stage, however, no clear mechanisms would be evident by the different drivers of mineral N concentrations between the regions. Nevertheless, land-use conditions would provide constraints on what critical soil variables drive the large-scale variation in $NH_4^+$ and $NO_3^-$ concentrations.

The Cubist models were able to account for 42–68% of the measured variation in both $NH_4^+$ and $NO_3^-$ in the soil across the sites and by each specific region (Table 3 and Fig. 4). The concentrations of $NH_4^+$ and $NO_3^-$ were reasonably estimated by the model on each region specific basis. In contrast, the model performance of soil mineral N was substantially limited due to high error, particularly with uncertainty in soil controls over a high range of concentrations at all sites and in the ecological region. In addition, overall model performance was possibly limited by the presence of small values so that any differences in the selected soil variables could not be fully considered in the model, eventually leading to the over-simplified controls in each of the regions. In this context, most of model errors resulted from the lack of accuracy mainly because the models were based on the limited data set and thus may not capture all the processes and resulting variation. Similar issues, such as the underestimation of relatively high $NO_3^-$ concentrations, have been found using model-based approaches (Johnsson et al., 1987; Wu and McGechan, 1998; Smith et al., 2008; Necpalova et al., 2015). The evaluation of these models suggests that the region-specific simulation of soil mineral N may not be sufficiently reliable at the continental scale, and this issue has been reported (Chang et al., 2001). These studies recommended further improvement in the mechanisms within the models, especially for hydrological or N transformation processes responsible for $NO_3^-$ concentrations. Specifically, we need to focus more on mechanistic roles of different soil drivers in N cycling between the regions, temporal controls over N transformation, and biogeochemical hot spots/moments in N retention and removal. As such, it is especially relevant to continue the development of accurate models in order to predict general patterns when integrating different land uses.





## 4.3 Soil stoichiometric controls on $NH_4^+$ and $NO_3^-$

Stoichiometric interactions of total soil C, N, and P can be geographically related to the transformation processes between soil organic and inorganic N over large spatial scales. The states of soil TOC, TN, and TP have been reported at the continental scale for Australia (Viscarra Rossel et al., 2015) and across Australia's major agro-ecological regions (Bui and Henderson, 2013). As far as we know, little or no studies have addressed the fate of soil mineral N by directly linking to soil stoichiometric interactions of C, N and P at the continental scale in Australia, because of practical difficulties to directly measure soil mineral N. We acknowledge this challenge and the need for more measurements at additional sites. Despite this limitation, we found that soil organic matter stocks and its $C:N:P$ stoichiometry were related to the potential to maintain or increase $NH_4^+$ concentrations in each region under different intensity of land uses. Specifically, there was a trend of increasing soil $NH_4^+$ concentrations with increasing $C:N$ (17.4 $\pm$ 9.9), $C:P$ (81.7 $\pm$ 55.2), and $N:P$ (4.6 $\pm$ 1.9) ratios (Fig. 5 and 6). There is also indication that the soil input stoichiometric ratios have less of an effect on final soil elemental ratios than previously expected, and depends mostly on soil mineralogy (Frossard et al., 2016). Similarly, Kirkby et al. (2011) found that the stable portion of organic matter had relatively constant $C:N:P$ ratios across a range of soil types. This suggests that soil $NH_4^+$ dynamics may depend on the amount of C and N stored in the soil as well as the relative P limitation in soil nutrient quality. The effects of these elemental interactions on soil $NO_3^-$ seemed to be important in the ecological region only (Fig. 5 and 6), specifically through changes in the amount of N and P retained in soil organic matter.

Different organic matter depletion and composition, and N mineralization from soil organic matter between the regions can limit the initial step for the terrestrial N cycle and determine the fate of mineral N (Booth et al., 2005; Gardenas et al., 2011; Denk et al., 2017). Recently, Tipping et al. (2016) reported global averages of the $C:N$ and $C:P$ stoichiometry, which approximately corresponded to 8.3 and 62.5 for nutrient-rich soil organic matter and 25.6 and 909.0 for nutrient-poor soil organic matter. The concentrations of C, N and P and corresponding elemental ratios in the soil suggest that TN was limited, while TP was abundant, in Australian soil relative to organic C. However, plant-available P is typically very low in Australian soil (Dalal et al., 2003; Viscarra Rossel and Bui, 2016). Soil $C:P$ and $N:P$ ratios are known to increase in response to disturbance events, such as the level of soil tillage and fire intensity (Orians and Milewski, 2007; Bui and Henderson, 2013). Our soil stoichiometry results, although limited, generally support the general status of nutrient-poor soil organic matter, as previously shown by Viscarra Rossel et al. (2015). More importantly, these regional patterns may not be consistent over different spatial scales. For example, Kirkby et al. (2011) reported significantly higher $C:P$ ratios for four Australian agricultural soils than international soils and thus potential P reduction due to total C loss at the field scale. Therefore, it is needed to estimate the importance of soil elemental interactions in determining the variation of mineral N at different spatial scales across and within various cropping and ecological conditions.

## 5 Conclusions

The distribution of soil $NH_4^+$ and $NO_3^-$ concentrations was significantly affected by regional differences of land use and management across Australia. Despite a wide range of soil conditions, the nature of the main soil controls and interactions





over soil N storage and availability appeared to differ across the sites and between the agricultural and ecological regions, constituting regionally explicit soil controls as subjected to different levels of human modification. Total organic C, TN, and TP had varying degrees of importance as a controlling factor between $NH_4^+$ and $NO_3^-$ and between regions. Assuming that total soil nutrients were related to the amount of soil N that can be retained, it was probably due to differences in the source type

5   of mineral N and under declining or steady-state soil organic N. In addition, more complex effects of biophysical properties on $NO_3^-$ than on $NH_4^+$ were found in both regions. Our results also suggest that stoichiometric interactions of C, N, and P may provide potentially important constraints to the dynamics of mineral N at the sites, specifically for $NH_4^+$ in each of the regions and $NO_3^-$ in the ecological region. Overall, large-scale $NH_4^+$ and $NO_3^-$ tended to be sensitive to N or P status relative to C in the soil nutrient budget, showing the biogeochemical role of soil nutrients in the regulation of soil mineral N cycling.

10  The Cubist model was effective at explaining the region-specific heterogeneity of Australian soils, empirically related to the concentrations of mineral N. However, the mechanisms of mineral N cycling controlled by the soil properties were not evident at the continental scale. In addition, the current data set and models still under-represent the intensive production and other agro-ecological zones of Australia, and therefore more focus should be given to a mechanistic understanding of the large-scale changes in soil mineral N retention and losses at this scale.

15  *Competing interests.*  The authors declare that they have no conflict of interest.

*Acknowledgements.*  We would like to acknowledge the contribution of the Biomes of Australian Soil Environments (BASE) consortium (https://data.bioplatforms.com/organization/pages/bpa-base/acknowledgements) in the generation of data used in this publication. The BASE project is supported by funding from Bioplatforms Australia through the Australian Government National Collaborative Research Infrastructure Strategy (NCRIS).



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

**Figure 1.** Location of 469 sampling sites across Australia, indicated with open circles for soils used for agricultural production and grazing (160 sites) and closed circles for conservation and natural environments (309 sites). The dark grey area represents intensive agricultural and plantation production. The light grey area represents production from relatively natural environments. The white area indicates the other ecological region.





**Figure 2.** Mineral N concentrations and fractions of total N in soil. Means between main agricultural and other ecological regions of Australia are significantly different at P-value < 0.001 (***), if indicated based on ANOVA on the log of the values. The horizontal line in the bar represents median. The error bar is the standard deviation of the mean.





**Figure 3.** Relative importance of soil attributes as the predictors of $NH_4^+$ and $NO_3^-$ concentrations (mg N kg$^{-1}$). The importance of the predictors is based on the usage of each variable in the rule conditions (grey bars) and in the Cubist model (black bars). Abbreviations: CEC, effective cation exchange; AWC, available water capacity.



**Figure 4.** Soil $NH_4^+$ and $NO_3^-$ concentrations on the log-transformed scale estimated by optimized Cubist models. Points represent model evaluation by 10-fold 50-repeated cross-validation. The error bar is the standard deviation of the estimated mean. The 1 : 1 line is indicated.



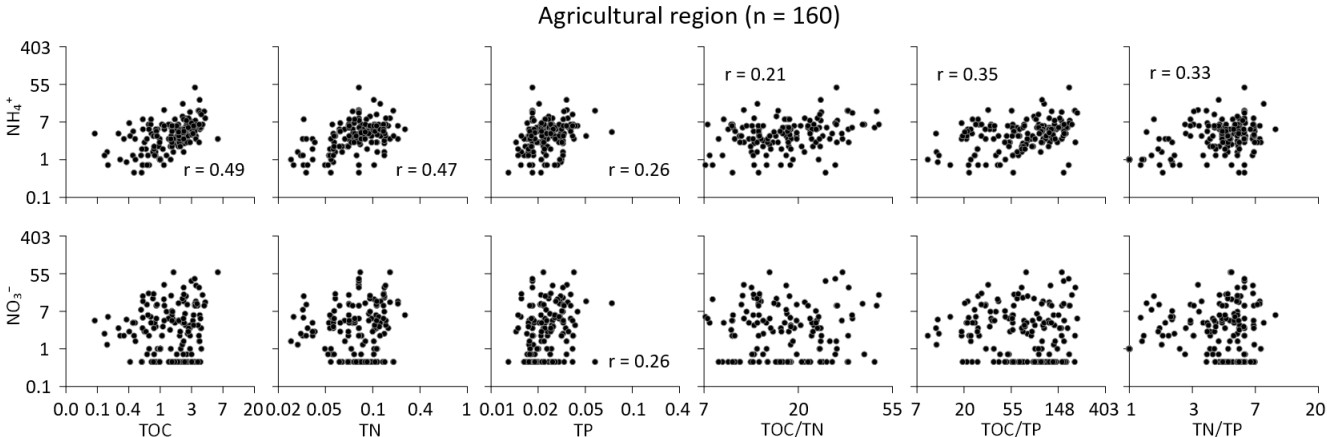

**Figure 5.** For $NH_4^+$ and $NO_3^-$ (mg N kg$^{-1}$), scatter plots of total organic C (TOC), total N (TN), total P (TP) and element ratios on the log-transformed scale in the agricultural region. The unit is expressed as percent C, N, or P by weight of soil. Pearson's correlation coefficient is reported only when the trend is significant (p < 0.05).

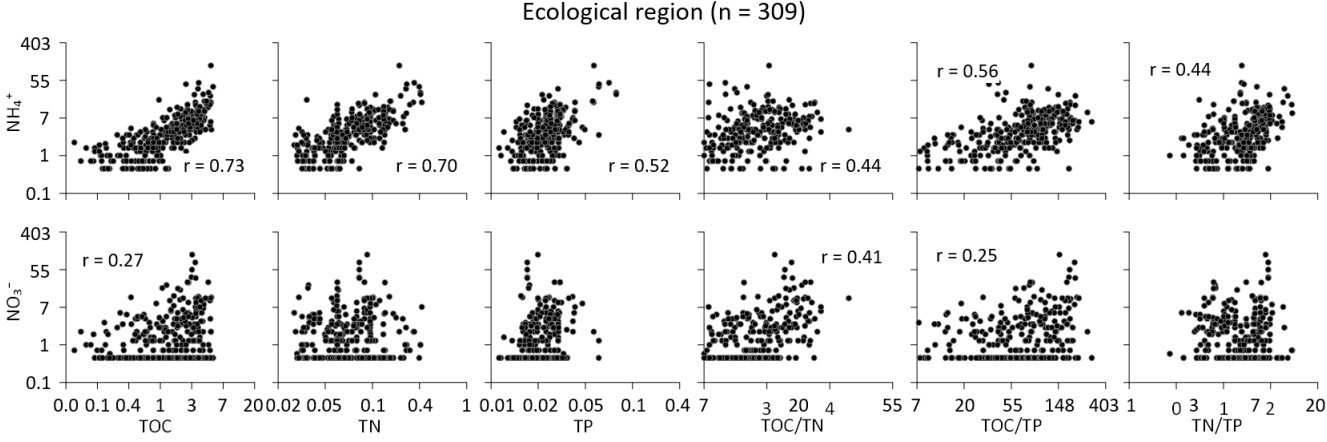

**Figure 6.** For $NH_4^+$ and $NO_3^-$ (mg N kg$^{-1}$),), scatter plots of total organic C (TOC), total N (TN), total P (TP) and element ratios on the log-transformed scale in the soil used for conservation and under natural vegetation. The unit is expressed as percent C, N, or P by weight of soil. Pearson's correlation coefficient is reported only when the trend is significant (p < 0.05).



**Table 1.** Summary of $NH_4^+$ and $NO_3^-$ concentrations (mg N $kg^{-1}$) in Australian soil by land use across 469 sites. Means with different letter are significantly different at $p < 0.05$, based on Tukey's HSD on the log of the values. Any categorical variables with a sample size of $< 10$ are considered in the multiple mean comparisons but not presented in the table. See Table S1 for the definition of land uses.

| | | $NH_4^+$ | | | | $NO_3^-$ | | | | |
|---|---|---|---|---|---|---|---|---|---|---|
| | n | Mean | | SD | Max | Min | Mean | | SD | Max | Min |
| Broad land use | | | | | | | | | | |
| Conservation and natural environments | 309 | 5.6 | | 9.0 | 120.0 | 0.5 | 3.8 | a | 9.9 | 121.0 | 0.5 |
| Production from dryland agriculture and plantations | 81 | 5.7 | | 6.1 | 46.5 | 0.5 | 7.8 | b | 11.3 | 59.0 | 0.5 |
| Production from irrigated agriculture and plantations | 11 | 6.1 | | 3.8 | 14.0 | 0.9 | 8.8 | b | 8.8 | 30.3 | 0.8 |
| Production from relatively natural environments | 68 | 3.7 | | 2.5 | 13.5 | 0.5 | 3.4 | a | 5.0 | 26.0 | 0.5 |
| | | | | | | | | | | |
| Detailed land use | | | | | | | | | | |
| Cropping | 16 | 3.4 | bc | 3.1 | 13.0 | 0.5 | 6.2 | ab | 5.3 | 18.5 | 0.5 |
| Environmental forest plantation | 12 | 4.8 | abc | 1.7 | 8.0 | 1.5 | 0.6 | cd | 0.2 | 1.3 | 0.5 |
| Grazing modified pastures | 45 | 7.3 | ac | 7.6 | 46.5 | 0.5 | 11.0 | a | 13.8 | 59.0 | 0.5 |
| Grazing native vegetation | 45 | 3.6 | bc | 2.3 | 11.0 | 0.5 | 4.8 | ab | 5.6 | 26.0 | 0.5 |
| Habitat/species management area | 10 | 28.1 | a | 37.0 | 120.0 | 2.0 | 1.3 | bcde | 0.8 | 2.5 | 0.5 |
| National park | 171 | 4.9 | bc | 5.4 | 39.0 | 0.5 | 3.3 | cde | 8.9 | 80.0 | 0.5 |
| Natural feature protection | 21 | 5.7 | abc | 3.6 | 12.0 | 0.8 | 9.0 | abe | 26.0 | 121.0 | 0.5 |
| Other conserved area | 18 | 3.1 | b | 4.0 | 15.0 | 0.5 | 2.9 | bcde | 3.5 | 11.0 | 0.5 |
| Production native forests | 23 | 4.0 | abc | 2.7 | 13.5 | 1.3 | 0.6 | d | 0.3 | 1.5 | 0.5 |
| Residual native cover | 60 | 5.3 | abc | 4.8 | 25.5 | 0.5 | 3.7 | b | 5.0 | 28.0 | 0.5 |

SD is the standard deviation of the mean.

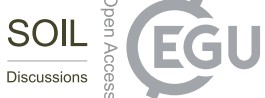


**Table 2.** Significant Pearson's correlation coefficients between soil $NH_4^+$ or $NO_3^-$ (mg N $kg^{-1}$) and selected soil variables at p-value < 0.05 (n = 469). The correlations indicated by "ns" are not significant.

| Variable | $NH_4^+$ | | | $NO_3^-$ | | |
|---|---|---|---|---|---|---|
| | All sites | Agricultural region | Ecological region | All sites | Agricultural region | Ecological region |
| Total organic C (%) | 0.67 | 0.49 | 0.73 | 0.21 | ns | 0.27 |
| Total N (%) | 0.65 | 0.47 | 0.70 | ns | ns | ns |
| Total P (%) | 0.46 | 0.26 | 0.52 | 0.14 | 0.26 | 0.08 |
| Sand (%) | -0.31 | -0.20 | -0.43 | -0.29 | -0.23 | -0.37 |
| Silt (%) | 0.41 | 0.17 | 0.53 | 0.24 | 0.16 | 0.3 |
| Clay (%) | 0.43 | 0.19 | 0.53 | 0.31 | 0.25 | 0.35 |
| Bulk density (g $cm^{-3}$) | -0.50 | -0.24 | -0.58 | ns | ns | ns |
| Cation exchange capacity (meq 100 $g^{-1}$) | 0.27 | ns | 0.33 | 0.51 | 0.52 | 0.50 |
| pH | -0.43 | -0.23 | -0.51 | 0.30 | 0.38 | 0.25 |
| Available water capacity | 0.12 | ns | 0.14 | ns | 0.35 | ns |

**Table 3.** Cross-validation statistics (mean $\pm$ standard deviation) of Cubist model on the estimation of $NH_4^+$ and $NO_3^-$ concentrations (mg N $kg^{-1}$) in soils. The performance of the models was evaluated with 10-fold 50-repeated cross-validation with instance-based corrections. The coefficient of determination ($R^2$), the root mean squared error (RMSE), estimated bias (ME), and the standard error of estimated bias (SDE) are considered.

| | | All sites | Agricultural region | Ecological region |
|---|---|---|---|---|
| $NH_4^+$ | $R^2$ | $0.60 \pm 0.11$ | $0.36 \pm 0.21$ | $0.68 \pm 0.10$ |
| | RMSE | $0.60 \pm 0.09$ | $0.65 \pm 0.16$ | $0.58 \pm 0.09$ |
| | ME | $-0.01 \pm 0.07$ | $-0.02 \pm 0.14$ | $-0.01 \pm 0.09$ |
| | SDE | $0.60 \pm 0.09$ | $0.66 \pm 0.16$ | $0.58 \pm 0.09$ |
| | | | | |
| $NO_3^-$ | $R^2$ | $0.42 \pm 0.13$ | $0.47 \pm 0.19$ | $0.47 \pm 0.16$ |
| | RMSE | $1.01 \pm 0.17$ | $0.99 \pm 0.22$ | $0.92 \pm 0.19$ |
| | ME | $0.00 \pm 0.13$ | $0.05 \pm 0.23$ | $0.05 \pm 0.16$ |
| | SDE | $1.01 \pm 0.18$ | $1.00 \pm 0.23$ | $0.92 \pm 0.19$ |