# Peer review of "Continental drivers of ammonium and nitrate in Australian soil under different land uses"

_SOIL, 2018_

## Referee Comment (RC1) · Anonymous Referee #1 · 14 May 2018

**Overview**

In this study the authors use a cubist model to decipher the soil parameters influential to mineral nitrogen (NH4+, NO3-) in Australian soils under different land-uses. The scientific question is valid and the data approach taken is state-of-the-art, so that the manuscript certainly falls within the scope of SOIL. However, the presentation and interpretation of the data lacks depth and specificity and needs significant revision before it can be accepted for publication.

**Comments**

**Abstract**

L5-6: It is unclear what the agricultural region and non-agricultural regions are. This

is a large problem for the paper, because samples taken from within the agricultural region may actually be from natural environments, or vice-verse.

L9: Normally NH4+ introduces H+ into the soil and therefore affects pH, not the other way around.

L11: what is the 'other' ecological region?

Introduction

P2, L1: Definition of mineral N?

P2, L6: I think it is the mineral N application, not continuous cropping per se, that leads to increased N emissions.

P2, L13: Mention some numbers/a range of expected mineral N values in soil.

P2L19-24: there are also studies showing that mineral N fertilizer can promote turnover of (and deplete) SOC (e.g. Shahbaz et al, LDD, 2017 or Neff et al., Nature, 2002).

P2, L34: Is biodiversity an ecosystem function? I would think it is an ecological indicator or even property, not a function.

Materials and methods

P3, L15-16: What is a unique environmental condition?

P3, L16-17: How was sampling performed? The text about no further samples being resourced is confusing.

P3 L17-22: 'Spatial distribution... at different times of year'. This section isn't methodology, it appears a combination of introductory remarks and discussion. Delete or move to the appropriate sections.

P4 L8-10: It is unclear whether samples from the 'agricultural region' really came from agriculturally used sites or were just located in what the authors appear to have defined to be a region dominated by agricultural use. Looking at the map, there are vast tracts

of land that have been defined as the agricultural production region which I would have thought are natural (e.g. nearly all the national parks in the great dividing range). How did you define these 'agricultural' and 'non-agricultural' regions? I am uneasy about such a broad sweeping definition being used to cover a continent (albeit a 'small' continent). If the samples are only from the agricultural region, does this mean that they were definitely taken from an agricultural land-use? This is a critical point concerning all the results.

P4, L13: briefly mention the depths again. Or state 'two sampling depths' or similar.

P4 L14, L20: I would have thought that a tree-based model such as cubist does not require log-transformation or the assumption of normality.

P4 L20, L23: Which 'selected' soil properties? A table with all the predictors used in the models would be nice.

P4 L27: A brief explanation of number of committees and nearest neighbours parameters would be helpful.

P4 L29: I dislike this use of the term relative importance. Your results show that several variable have a relative importance of 100 % in the same model. This implies that total importance is > 100 %. Relative importance in other models is frequently based upon influence of the predictor on model accuracy/goodness-of-fit, not solely on its inclusion in the model. Perhaps just use 'importance'?

Results

P5 L13: 'regional patterns' I do not think this is an appropriate use of 'regional'. Normally, a region is an area which is spatially defined because it is smaller and belongs together (e.g. Gippsland, the Hunter Valley, The West Australian Wheat belt). Looking at Figure 1, both the ecological and the agricultural production regions cover disjoint areas which have completely different climates, geology, vegetation among other things. I do not feel comfortable clumping these 'regions' together unless the samples in the

regions are really defined by land-use, in which case you should refer to land-use categories, not regions.

P5 L15: 'or the median of' - rephrase.

P5 L19: 'significant regional difference' - be specific.

P5 L21-22: Do you mean that NH4 and NO3 differed between land-uses?

P5 L23-24: 'relatively natural environments' - What specific land-uses or environments are you referring to?

P5 L24-25: But the table does show e.g. significantly higher NH4 in 'habitat/species management'. Make sure the text fits the results you present.

P5 L30: Which soil properties, what were the effects? Be specific.P6 L1: Once again 'soil properties'. This is too vague as to be meaningful., Be specific.

P6 L28-31: This is discussion, not results.

Discussion

P7 L9: 'may suggest' - or just suggests? No need to be so hesitant to make a statement.

P7 L14-15: Now you are talking about land-uses, not regions. Please be consistent. What about the effects of different climate and geology. These have a massive influence on soil forming factors, as well as vegetation. I would have thought that these factors could be accounted for in models. Even if they haven't, they should be given some thought in the discussion.

P7 L16: This seems to be presentation of new results in the discussion section.

P7 L18-19: This is the crux of the issue with this paper - it is unclear how you defined agricultural vs. ecological regions, so you are not really comparing land-use effects. In fact, it remains unclear to me what you a comparing, given the very large areas covered

by the 'agricultural' and 'ecological' regions, which cover vastly differing climates and site environments.

P7 L26: What is a soil disturbance level?

P8 L18: I find this explanation for the lack of climate consideration inadequate. You can easily download at least broad climate data from the BOM and could have considered this in the models. You have clumped sites from Tasmania together with sites from far-north QLD, which have vastly differing climates. Even something as broad as a Köppen climate classification may have been considered in your models, if you do not have access to something more specific.

P8 L30: Sequentially?

P8 L31: Do not give examples. You should discuss the actual results.

P9 L4: Do not give one example, discuss your findings.

P9 L5: The model explained variance was much lower for NH4+ in the agricultural soil, potentially indicating that you have not included all the driving factors in your models.

P9 L5-10: Which effects - specifically state what you think the relationship is? See comment above on relationship between pH and NH4+. Make sure you discuss your results, not just reiterate them.

P9 L10-13: 'Was affected by' - this is too vague. Be specific. What was the relationship? Positive or negative? Are you sure this is causation, or is it merely covariance...?

P9 L24-25: What about the error arising from a lack of consideration of other factors driving soil processes (climate, geology, topography...)?

P10 L9: Where is land-use intensity presented?

Conclusions

I do not think you actually looked at land use and management in depth (except the

results presented in Table 1, which for NH4 appear insignificant across the broad categories). You defined agricultural and ecological zones, but the way this was done is unclear.

P10 L34 (35?): For me, your results do not indicate regionally explicit soil controls, but I find your definition of these regions problematic.

P11 L 4: 'it was probably due to' What are you referring to with it?

P11 L5: which complex biophysical properties?

Tables and Figures

Figure 1: Throughout the manuscript you contrast the agricultural region with the ecological region, but the map shows three regions 'intensive agricultural and plantation production', 'production from relatively natural environments' 'other ecological region'. It is unclear how you have defined your regions.

Figure 2: I think a box-whisker plot would be much better here.

Supplementary Figure 1: You state on P4 L27 that the number of committees was set to one to avoid complex models. Why does the y-axis show up to 20 committees? The caption mentions grey bars, but there do not appear to be any. Supplementary Figure 2: You state on P4 L27 that the number of committees was set to one to avoid complex models. Why does the y-axis show up to 20 committees?

---

## Referee Comment (RC2) · Anonymous Referee #2 · 20 Jun 2018

The paper uses a large data set of soil ammonium and nitrate concentrations and attempts to correlate these values with various soil properties and with land use. Although the paper does not represent novel concepts, it uses a state of the art analysis and a large data set. Given the size of the surface under investigation, the data set is however relatively sparse (as acknowledged by the authors). Nonetheless, the study represents a contribution to scientific progress and an important basis for the investigation of large-scale drivers of soil N, thus in my opinion warranting publication. I have however a couple of main concerns with the paper, which should be addressed before publication. The analyses are clearly outlined and assumptions seem to be valid (with one exception, as noted below). The paper is well structured and generally well written, although there are several cases where the text is unclear; these exceptions are outlined below. Additionally, the conclusion section contains some terms that are not addressed in the paper and I have suggested their removal (see below).

Main comments

I have two main concerns with the manuscript.

**Firstly**, the authors state that they are investigating drivers of soil ammonium and nitrate. This is indicated by the language used throughout the study ('controls' and 'drivers'), including in the aims section. The study is however an observational study, meaning cause and effect cannot be derived from these results; the 'drivers' of soil ammonium and nitrate cannot be identified from such a study, only correlated variables (or 'patterns'). It would be acceptable to state that this study aims to identify candidate soil properties that might be considered, following further study, as 'controls' or 'drivers'; the study cannot however identify these controls and drivers itself. Such wording would agree with the fact that this study represents a basis for further studies –as indeed the authors state several times.

The text throughout the manuscript (including the title) needs to be corrected to reflect this. Words such as 'controls' and 'drivers' need to be avoided.

**Secondly**, it is unclear to me why the authors have split the soil samples into the two regions ('agricultural' and 'ecological'), given that within the agricultural region there seems to be a large variation in the intensity of land management:

- Would it not make more sense to use the actual land use of the 469 sites as a factor, or to derive a scale of land management intensity from the 28 land use types, and examine the correlation of this with the soil attributes?
- The authors state that they aim to assess the relative importance of large-scale drivers, which I assume is the purpose of using two large regions (agriculture and ecological). However, I suspect these two regions are inadequate to do this: given that the two regions encompass very large climatic and geological variation, important large-scale potential 'drivers' are not addressed by this method either. The authors could consider incorporating broad climatic information into the analysis. This may reduce some of the noise and thus improve the outcome of the analyses.

I have a number of additional minor concerns with the paper that need to be addressed:

P3 L15-16: I appreciate that the sampling design of the BASE project is described elsewhere, but it would be useful to have a little more detail on this, including the number of soil samples taken from each 25m x 25m 'site'.

P3 L 27-33: The soil maps used were a result of spatial modelling. The outcome of a model cannot be considered as data; please therefore change the word 'data' (L27 and L32) to 'values' or 'information'.

P5 L8-10: The values given in these lines (sum of NH4+ and NO3-, NH4+ and NO3-) seem to refer to mean values (for the first of this set of values, it is indeed stated so). Would it not make more sense to give median values here, given that median values are what are shown on the corresponding graphs (figure 2)? The mean average value of a population that is not normally distributed is not particularly informative. Additionally, given that the NH4+ and NO3- concentrations are not normally distributed, stating the standard deviation of these data is misleading, as the use of a SD value to convey information assumes the population is more or less normally distributed.

P5 L 19-13: Why carry out an analysis omitting the samples for which NO3- concentration was below the detection rate? If a soil sample has a NO3- concentration below the detection level, this does not equate with 'no data', but rather means that the NO3- concentration is simply very low (as the authors indeed assume). Unless something specific is being tested, which I do not think is the case here, an analysis with these points removed is uninformative.

This subject is re-visited in the discussion (P7 L11-12) but as the text is written, I still do not understand what information this analysis brings. The results of this extra analysis do however indicate that the agricultural soils have a bimodal distribution with respect to NO3- concentration, i.e. many soils have very low concentration and many soils have a very high concentration (indicative of high addition rates of NO3-). If this additional analysis was carried out to illustrate this, the authors need to make this clear in the discussion, and indeed expand this point in the discussion.

P7 L18-19: What is meant here by this sentence, particularly the term "agricultural soil"? Do the authors mean that not every site in the agricultural region is under agriculture? If so, please change accordingly and change the terminology to 'soil under agriculture'. If not, please explain the term 'agricultural soil'.

P8 L2-3: This sentence is either incorrect (as I have understood it) or imprecisely written: I understand from this sentence that the NH4+ concentrations between the soils from the agricultural and ecological regions are different. According to figure 2 and text in the results section (P5 L8-9) however, NH4+ concentrations are similar. I suspect a more complex pattern is meant by the authors; this needs to be more clearly explained.

P9 L 18-25: The text here is difficult to understand. I have a few suggestions that might help:

L 19: Replace "by each specific region" by "within each region" (if I have understood correctly).

L20: Replace "on each region specific basis" with 'for each region" (if I have understood correctly).

The sentence L20-22 is very unclear. Is it referring to the higher prediction error for the high concentrations of NO3- (in the ecological region in particular)? Rewrite.

L23: "presence of small values" is too vague. Do the authors mean that the high frequency of samples with very small NO3- concentrations is the cause of the limited overall model performance? Needs to be explained.

L24: Replace "most of model errors" with "much of the model error"

L24-26: This sentence is too vague. This sentence relates to the presence of small values of What is meant be "the limited data set" exactly?

P10 L7-9: The authors here imply that they have identified a process in the results, the "potential to maintain or increase $NH_4^+$ concentrations". This process is a possible explanation of the results they have found, but is not in itself a result. Please change text accordingly.

P10 L33-34: The term "management" should be avoided here, unless the authors specify what they mean by management, as a term separate from and in addition to 'land use'.

P 11 L 2: The term "human modification" should be removed; this term implies some sort of scale or land use intensity (e.g. nutrient input levels), but this has not explicitly been investigated in this study.

Figure 1: Three main regions are shown here, whereas two are considered in the text. I recommend that the number of regions considered should be consistent. Alternatively, if the three ecological regions were considered distinct enough to warrant their separation on the map, why not use three regions in the analysis?

Technical corrections

P1 L11: It is unclear what the 'other' ecological region refers to.

P3 L17-22: The sentences from L17 to 22 need to be moved out of this section; I suggest to the discussion.

P5 L5-6: In the first sentence of the results, the mean $NH_4^+$ and $NO_3^-$ concentrations are stated, referring to figure 2. However, in figure 2, the median concentrations are given. Please correct text (or change figure 2) accordingly.

P5 L25: Change "large" to "high"

P5 L26: "in that" needs to be inserted between the words "…..environments or" and "used mainly…..".

P6 L16-17: This sentence belongs in the discussion.

P7 L8: Remove "In our case".

P7 L7: Change "which may suggest" to "which suggests"

P7 L22-23: Change "…which was considerably low in the soil from the agricultural region compared to the ecological region" to "… which was considerably lower in the soil from the agricultural region compared to that in the ecological region".

P10 L6: Change "difficulties to directly measure" to "difficulties in directly measuring"

P10 L10: Change "also indication that' to 'also an indication that"

P10 L11: Unclear. What depends mostly on soil mineralogy? Soil input stoichiometric ratios or final soil elemental ratios?

P10 L28-29: Unclear. I think this sentence needs to be re-written as: "Therefore, the importance of soil elemental interactions in determining the variation of mineral N at different spatial scales across and within various cropping and ecological conditions needs to be estimated." I have possibly interpreted this sentence incorrectly. If this is the case, what is "it" (L28)?

Figure 2: Please clarify, do the error bars shown represent the standard deviation of the data values i.e. population, or the standard error of the mean?

Figure 4: Is 'standard error of the (estimated) mean' meant here?

---

## Author Comment (AC1) · 6 Jul 2018

Overview

In this study the authors use a cubist model to decipher the soil parameters influential to mineral nitrogen ($NH_4^+$, $NO_3^-$) in Australian soils under different land-uses. The scientific question is valid and the data approach taken is state-of-the-art, so that the manuscript certainly falls within the scope of SOIL. However, the presentation and interpretation of the data lacks depth and specificity and needs significant revision before it can be accepted for publication.

We have addressed all the comments carefully to improve the overall manuscript. Please refer to our responses below.

Comments

Abstract

L5-6: It is unclear what the agricultural region and non-agricultural regions are. This is a large problem for the paper, because samples taken from within the agricultural region may actually be from natural environments, or vice-verse.

Thank you for pointing this out. Now the "ecological" region has been renamed as the "non-agricultural" region throughout the manuscript. Please note that the sample locations were classified by the broad and detailed land uses from the Australian Bureau of Agricultural and Resource Economics and Sciences (ABARES) land use map of 2016, as shown in Figure 1 and Table S1, which were grouped into two regions. However, you are correct that some of the samples within the agricultural production areas are from a natural environment, and vice versa. Nevertheless, given the large amount of samples in this data set, we are confident that the vast majority of samples from either region are accurately described by the given grouping (Table S1). Furthermore, despite these potential issues, which we discuss in the discussion section (P7 L18-19), we believe these general classifications are able to adequately show differences between the two broad land-use type across the continent.

L9: Normally $NH_4^+$ introduces $H^+$ into the soil and therefore affects pH, not the other way around.

$NH_4^+$ concentrations are also affected by pH, particularly during microbial nitrification (e.g., Cuhel et al. Appl. Environ. Microbiol. 2010 vol. 76 no. 6 1870-1878). In this study, we focused on the role of pH as a potential soil control.

L11: what is the 'other' ecological region?

Please see previous response -- we changed this region to the non-agricultural region to avoid confusion.

Introduction

P2, L1: Definition of mineral N?

The definition of mineral N has been added. The sentence now reads: "In addition, different mineral forms of N **(e.g., $NH_4^+$ and $NO_3^-$)** often …"

P2, L6: I think it is the mineral N application, not continuous cropping per se that leads to increased N emissions.

We agree that a level of fertiliser N application is closely related to N gaseous emissions. In addition, diversifying the sources of N fertilization, such as N fixed by legumes, may also result in some emissions from cropping systems. Nonetheless, we have modified the sentence and added that emissions are "**mainly through N fertilizer application**".

P2, L13: Mention some numbers/a range of expected mineral N values in soil.

We have checked the references. Viscarra Rossel and Bouma (Agric. Sys. 2016 vol. 148 71–74) estimated $NO_3^-$ contents in the range of 0–200 mg N $kg^{-1}$ for cropping systems, based on the data from www.bfdc.com.au. Unfortunately, we were not able to find a good reference on the regional/continental scale that is compatible with our study. To point out this lack of information, we have added a sentence: "**However, such large-scale soil information is generally not available.**"

P2L19-24: there are also studies showing that mineral N fertilizer can promote turnover of (and deplete) SOC (e.g. Shahbaz et al, LDD, 2017 or Neff et al., Nature, 2002).

Thank you for the references. We have added a sentence to address a decrease in SOC by added N "**Conversely, there are studies showing that SOC can be depleted by N fertilizer application (Neff et al., 2002; Shahbaz et al., 2017)**".

P2, L34: Is biodiversity an ecosystem function? I would think it is an ecological indicator or even property, not a function.

Thank you for picking this up. We agree that biodiversity is not an ecosystem function. We simply intended to infer the status of biodiversity is related to ecosystem functioning. Therefore, we changed "ecosystem functions" to "**ecological properties**" in the sentence.

Materials and methods

P3, L15-16: What is a unique environmental condition?

Please refer to our response below.

P3, L16-17: How was sampling performed? The text about no further samples being resourced is confusing.

The BASE project performed the soil sampling, and one of their criteria was to choose sampling location across a wide range of environmental conditions and unique combinations of soil, climate and management. To make it clear, we have modified the sentence and added the following short description of soil sampling: "**Each sample was collected from a site that represented a unique combination of soil, climate and management. Specifically, between 9 and 30 soil cores were sampled in a 25 m x 25 m quadrat and split into two different depths, 0–0.1 m and 0.2–0.3 m, respectively, and then combined into one composite sample (approximately 1 kg of soil) for each depth.**"

Please note that we have deleted "because no further samples were resourced". This simply means that the BASE project had only sampled once and the project ended.

P3 L17-22: 'Spatial distribution. . . at different times of year'. This section isn't methodology, it appears a combination of introductory remarks and discussion. Delete or move to the appropriate sections.

This was one of the important assumptions for our modelling approach so we would rather consider this as a method. One of the common known issues to study large-scale soil mineral N dynamics is a difficulty in representing the study over time because of its temporally variable dynamics. To address this comment, we have deleted the sentence (previously on L22), "However, this source of uncertainty …"

P4 L8-10: It is unclear whether samples from the 'agricultural region' really came from agriculturally used sites or were just located in what the authors appear to have defined to be a region dominated by agricultural use. Looking at the map, there are vast tracts of land that have been defined as the agricultural production region which I would have thought are natural (e.g. nearly all the national parks in the great dividing range). How did you define these 'agricultural' and 'non-agricultural' regions? I am uneasy about such a broad sweeping definition being used to cover

a continent (albeit a 'small' continent). If the samples are only from the agricultural region, does this mean that they were definitely taken from an agricultural land-use? This is a critical point concerning all the results.

We apologise for the confusion. Now the two regions are re-named as "agricultural" and "non-agricultural" to exactly match broad land use labels from the ABARES land use map, published in 2016 (see Supplementary Table 1). Samples from the production region solely came from sites used for any agricultural production activity – i.e., dryland and irrigated cropping and improved and native pastures used for animal grazing – see Table S1. The other samples were collected from sites under conservation and in natural environment, such as national park and residual native cover.

In our revision, we re-defined these regions as: "**(2) the samples from the sites that originate from dryland and irrigated cropping, and from improved and native pastures used for animal grazing (hereafter referred to as the "agricultural" region), and (3) the samples from the sites that were conserved and in natural environments outside of the agricultural production zones (referred as the "non-agricultural" region)**". We believe that these terms are clearer.

The caption of Figure 1 has been changed to: "**Location of 469 sampling sites across Australia, indicated with open circles for soils used for dryland and irrigated cropping, and from improved and native pastures used for animal grazing (160 sites) and closed circles for soils from areas that are conserved and in natural environments outside of the agricultural production zones (309 sites). The dark grey area represents intensive agricultural and plantation production. The light grey area represents agricultural production from relatively natural environments. The white area indicates the non-agricultural region.**"

P4, L13: briefly mention the depths again. Or state 'two sampling depths' or similar.

Thank you. We have stated **'two sampling depths'** as suggested.

P4 L14, L20: I would have thought that a tree-based model such as cubist does not require log-transformation or the assumption of normality.

This is indeed correct. However, a tree-based model consists of linear regression models, which would be more robust when the values are log-transformed.

P4 L20, L23: Which 'selected' soil properties? A table with all the predictors used in the models would be nice.

Thanks for the comment. All the soil properties used in the models are listed in the section 2.1. Nevertheless, it may be a good idea to repeat the list again. We have revised the sentence to, "**The concentrations of soil $NH_4^+$, $NO_3^-$, TOC, TN, TP, CEC, and the fraction of sand, silt, and clay, BD, and AWC, except for pH, were …**"

P4 L27: A brief explanation of number of committees and nearest neighbours parameters would be helpful.

We fixed the number of committees to one, although we have tested and reported the importance of soil controls in more complex models in supplementary figures (with 5, 10 and 20 committees). We have added "**with 3-9 neighbors**" to the sentence, "The optimized models ...".

P4 L29: I dislike this use of the term relative importance. Your results show that several variable have a relative importance of 100 % in the same model. This implies that total importance is > 100 %. Relative importance in other models is frequently based upon influence of the predictor on model accuracy/goodness-of-fit, not solely on its inclusion in the model. Perhaps just use 'importance'?

We fully agree and thus have corrected the term as suggested throughout the manuscript. Perhaps, we may have to re-define the term importance of variables from Cubist modeling. Please note that this is how 'relative importance' is currently defined without a full consideration of model coefficients.

Results

P5 L13: 'regional patterns' I do not think this is an appropriate use of 'regional'. Normally, a region is an area which is spatially defined because it is smaller and belongs together (e.g. Gippsland, the Hunter Valley, The West Australian Wheat belt). Looking at Figure 1, both the ecological and the agricultural production regions cover disjoint areas which have completely different climates, geology, vegetation among other things. I do not feel comfortable

clumping these 'regions' together unless the samples in the regions are really defined by land-use, in which case you should refer to land-use categories, not regions.

We hope we have addressed this concern by re-naming the regions according to broad land uses. All the samples in each of the regions are strictly defined by land use.

P5 L15: 'or the median of' - rephrase.

Based on the comment from the other reviewer, we have reported the median values instead of the mean values. Exception was when we reported mean comparisons. So, this has been deleted.

P5 L19: 'significant regional difference' - be specific.

This has been corrected to "**a significantly higher $NO_3^-$ fraction of TN in the agricultural region than in the non-agricultural region.**"

P5 L21-22: Do you mean that $NH_4$ and $NO_3$ differed between land-uses?

Yes, that is correct. We clarified the sentence to: "**The contents of $NH_4^+$ and $NO_3^-$ differed by both broad and detailed land uses (Table 1).**".

P5 L23-24: 'relatively natural environments' - What specific land-uses or environments are you referring to?

We hope that this has been resolved with our new terms with the new definitions.

P5 L24-25: But the table does show e.g. significantly higher $NH_4$ in 'habitat/species management'. Make sure the text fits the results you present.

For this land use category, the concentration of $NH_4^+$ appeared to be high among the detailed land uses. However, it was not significantly higher except for cropping, grazing native vegetation, national park, and other conserved areas. We revised and corrected the text from "little or no differences" to "**no apparent differences**".

P5 L30: Which soil properties, what were the effects? Be specific.

These soil properties were listed as "**TOC, TN, TP, the clay fraction, CEC and pH**". We deleted ", except the sand and silt fractions, BD and AWC" to be specific as suggested.

P6 L1: Once again 'soil properties'. This is too vague as to be meaningful. Be specific.

We revised the sentence from "by soil properties, similar to the controls on soil $NH_4^+$" to "**by similar soil controls as for $NH_4^+$**".

P6 L28-31: This is discussion, not results.

Thank you. We agree and we have deleted this sentence form the results.

Discussion

P7 L9: 'may suggest' - or just suggests? No need to be so hesitant to make a statement.

Thank you. We were being conservative in the interpretation of our results because of the limitations of the data, but we agree and have now used 'suggests'.

P7 L14-15: Now you are talking about land-uses, not regions. Please be consistent.
What about the effects of different climate and geology. These have a massive influence on soil forming factors, as well as vegetation. I would have thought that these factors could be accounted for in models. Even if they haven't, they should be given some thought in the discussion.

We revised the sentence "under different land-use conditions" to "**in different regions defined by broad land uses**" for consistency.

We agree with the reviewer's general point about different environmental factors, however, as described in section 2.1, the data from the soil maps, which we used, already accounted for climate, mineralogy, etc. because these covariates were used to derive the soil maps.

P7 L16: This seems to be presentation of new results in the discussion section.

These are not new results. To make this clear, we have revised the sentence to "**We found complex, but consistent regional patterns of soil NH$_4^+$ and NO$_3^-$ by broad land uses.**"

P7 L18-19: This is the crux of the issue with this paper - it is unclear how you defined agricultural vs. ecological regions, so you are not really comparing land-use effects. In fact, it remains unclear to me what you a comparing, given the very large areas covered by the 'agricultural' and 'ecological' regions, which cover vastly differing climates and site environments.

We hope that our revision of the definition of these regions has been resolved this misunderstanding.

P7 L26: What is a soil disturbance level?

We have revised "at various levels" to "**and various tillage intensities**".

P8 L18: I find this explanation for the lack of climate consideration inadequate. You can easily download at least broad climate data from the BOM and could have considered this in the models. You have clumped sites from Tasmania together with sites from far-north QLD, which have vastly differing climates. Even something as broad as a Köppen climate classification may have been considered in your models, if you do not have access to something more specific.

We disagree with this comment. We know that the continental climate data are available, and we have them and used them to derive the soil maps in a previous work, Viscarra Rossel et al (2015). Here, we are referring to the current BASE data. Please also note that we mentioned that "most of the sites were located in arid or temperate ecological zones". We agree that effects of climate are potentially influential for soil mineral N distribution but probably as distal controls (not as proximal controls). If we had access to soil-water or soil temperature data, this would be definitely be something to consider in our modelling – but we do not.

P8 L30: Sequentially?

We have deleted it.

P8 L31: Do not give examples. You should discuss the actual results.

We have changed "For example," to "**It is well known that**".

P9 L4: Do not give one example, discuss your findings.

We have removed "(e.g., TP)" as you suggest, and from our point of view have adequately discussed the results based on the objectives of the paper. Here we highlight different soil controls in different regions or by dominant land uses at the regional scale.

P9 L5: The model explained variance was much lower for NH$_4^+$ in the agricultural soil, potentially indicating that you have not included all the driving factors in your models.

Having more variables in a model might increases a proportion of explained variation, if these are related to the response. However, the lack of explained variance may also indicate that NH$_4^+$ in agricultural soil is highly variable and at much shorter distances, based on a sample variogram calculated from the data (results not shown). Unfortunately, we were not able to assess whether the problem was with the spatial variability as we had not enough data to derive models of the spatial variation (i.e., variograms).

P9 L5-10: Which effects - specifically state what you think the relationship is? See comment above on relationship between pH and NH$_4^+$. Make sure you discuss your results, not just reiterate them.

We have added "**positively**" to show how $NH_4^+$ was related to TN. We have discussed our results whenever reasonably allowed at the regional scale (see on L8-10).

P9 L10-13: 'Was affected by' - this is too vague. Be specific. What was the relationship? Positive or negative? Are you sure this is causation, or is it merely covariance. . .?

We have added "**positively**". For sure, we 'assume' causation in our modelling approach.

P9 L24-25: What about the error arising from a lack of consideration of other factors driving soil processes (climate, geology, topography. . .)?

The factors mentioned by the reviewer and those not mentioned are all potentially important. Please note that soil maps provide an integrated measure of climate, geology, terrain, etc. Here, we have not directly but indirectly accounted for these factors. Please also note that our main objective was to determine continental/regional soil factors. Thus, this point of discussion is out of our current scope.

P10 L9: Where is land-use intensity presented?

We have deleted "intensity of".

Conclusions

I do not think you actually looked at land use and management in depth (except the results presented in Table 1, which for $NH_4$ appear insignificant across the broad categories). You defined agricultural and ecological zones, but the way this was done is unclear.

All comparisons are based on broad land uses, not so much on detailed land uses. As shown in Supplementary Table 1, we were not able to address at the level of each of the specific land use types.

P10 L34 (35?): For me, your results do not indicate regionally explicit soil controls, but I find your definition of these regions problematic.

In this study the goal was not to understand differences in different geographic regions, but instead under different land-uses (i.e. agricultural versus non-agricultural) across the continent. Hopefully this is more clear given our new terminology.

P11 L4: 'it was probably due to' What are you referring to with it?

Here, "It" was clarified as: "**the effects of total soil elements**".

P11 L5: which complex biophysical properties?

We have modified "biophysical properties" to "**soil properties**".

Tables and Figures

Figure 1: Throughout the manuscript you contrast the agricultural region with the ecological region, but the map shows three regions 'intensive agricultural and plantation production', 'production from relatively natural environments' 'other ecological region'. It is unclear how you have defined your regions.

The caption now include the definition of the "agricultural" and "non-agricultural" regions. These regions have been re-defined in the section 2.2.

Figure 2: I think a box-whisker plot would be much better here.

We have made a new figure as suggested.

Supplementary Figure 1: You state on P4 L27 that the number of committees was set to one to avoid complex models. Why does the y-axis show up to 20 committees? The caption mentions grey bars, but there do not appear to be any.

For this figure, the caption has been changed as: "**Supplementary Figure 1. Importance of soil properties as the predictors of NH$_4^+$ contents (mg N kg$^{-1}$). The importance of the predictors is based on the usage of each variable in the Cubist model (black bars). None of soil properties is used to set the rule conditions.**"

There is no grey bar because none of the soil properties was used to set model rules. We reported the results from model with only one committee. Please note that this is the supplementary information, as described in the section 2.1 on P4 L31.

Supplementary Figure 2: You state on P4 L27 that the number of committees was set to one to avoid complex models. Why does the y-axis show up to 20 committees?

Please refer to our response about the comment on Supplementary Figure 1.

Please note that we have made a new Table 1.

Additional technical corrections:

We have added the units of measured soil properties of the BASE data and values from the soil maps in the section 2.1. NH$_4^+$ and NO$_3^-$ were reported in mg N/kg. Organic C, total N, total P, and texture were reported in %. Bulk density and exchangeable CEC were reported in g/cm$^3$ and meq/100 g, respectively. Thus, we have corrected the term "concentration" to "content" for soil mineral N as determined by the mass of sample throughout the manuscript.

---

## Author Comment (AC2) · 6 Jul 2018

The paper uses a large data set of soil ammonium and nitrate concentrations and attempts to correlate these values with various soil properties and with land use. Although the paper does not represent novel concepts, it uses a state of the art analysis and a large data set. Given the size of the surface under investigation, the data set is however relatively sparse (as acknowledged by the authors). Nonetheless, the study represents a contribution to scientific progress and an important basis for the investigation of large-scale drivers of soil N, thus in my opinion warranting publication. I have however a couple of main concerns with the paper, which should be addressed before publication. The analyses are clearly outlined and assumptions seem to be valid (with one exception, as noted below). The paper is well structured and generally well written, although there are several cases where the text is unclear; these exceptions are outlined below. Additionally, the conclusion section contains some terms that are not addressed in the paper and I have suggested their removal (see below).

Thank you for the positive comment. We acknowledge that the BASE data is sparse with only 469 sites over all of Australia. But as far as we know, the data is the largest dataset available (possibly worldwide) to study the spatial distribution of soil mineral N and its soil controls.

Main comments

I have two main concerns with the manuscript.

**Firstly**, the authors state that they are investigating drivers of soil ammonium and nitrate. This is indicated by the language used throughout the study ('controls' and 'drivers'), including in the aims section. The study is however an observational study, meaning cause and effect cannot be derived from these results; the 'drivers' of soil ammonium and nitrate cannot be identified from such a study, only correlated variables (or 'patterns'). It would be acceptable to state that this study aims to identify candidate soil properties that might be considered, following further study, as 'controls' or 'drivers'; the study cannot however identify these controls and drivers itself. Such wording would agree with the fact that this study represents a basis for further studies –as indeed the authors state several times. The text throughout the manuscript (including the title) needs to be corrected to reflect this. Words such as 'controls' and 'drivers' need to be avoided.

The BASE data include soil $NH_4^+$ and $NO_3^-$ as well as soil variables at the continental scale. The reviewer is correct in a sense that we cannot strictly identify direct soil controls on mineral N and imply any mechanistic relationships based on the data. Nevertheless, modelling with Cubist assumes that the independent variables are state-factors that affect soil mineral N, and therefore may be assumed to be 'controls'. We agree that we need to be conservative with our interpretations of the effects of the independent variables on soil mineral N because of the sparseness of the data, and we believe that we have been and that we have clearly stated our assumptions . Further, we suggest that even in small-scale manipulative studies, it is often not straightforward to derive cause and effect relationships in soil system without making assumptions.

**Secondly**, it is unclear to me why the authors have split the soil samples into the two regions ('agricultural' and 'ecological'), given that within the agricultural region there seems to be a large variation in the intensity of land management:

- Would it not make more sense to use the actual land use of the 469 sites as a factor, or to derive a scale of land management intensity from the 28 land use types, and examine the correlation of this with the soil attributes?

  Thank you for this suggestion. We split the data into the two regions because the data contain only 469 sites which do not sufficiently represent Australia at the detailed land uses level. In addition, detailed land uses are not well (and equally) balanced in the data (see Table S1). Thus, our regional modeling by accounting broad land uses is the only feasible approach to meet our objectives. Using land use classes as a factor in the modelling will not help to improve our interpretations or help to address our aims. Furthermore, we unfortunately do not have any additional, and more importantly necessary, supplementary information on management practices which could be used to derive a land-use intensity scale across the sites. We can see that this option may be feasible with fully resolved spatial data or a well-established ecosystem model, not an empirical model.

- The authors state that they aim to assess the relative importance of large-scale drivers, which I assume is the purpose of using two large regions (agriculture and ecological). However, I suspect these two regions are inadequate to do this: given that the two regions encompass very large climatic and geological variation, important large-scale potential 'drivers' are not addressed by this method either. The authors

could consider incorporating broad climatic information into the analysis. This may reduce some of the noise and thus improve the outcome of the analyses.

We disagree with this comment. For the study, the soil property values from our recent mapping were generated by considering full continental variation in climate and other environmental variables, as stated in P3 L33 and P4 L1 of our manuscrip. In this sense, the soil data that we used 'integrates' information from climate and the other environmental variables that determine the distribution of the particular soil properties at the continental scale. In this study, our focus was on proximal soil controls on mineral N, rather than distal controls, such as air temperature or rainfall.

I have a number of additional minor concerns with the paper that need to be addressed:
P3 L15-16: I appreciate that the sampling design of the BASE project is described elsewhere, but it would be useful to have a little more detail on this, including the number of soil samples taken from each 25m x 25m 'site'.

We have added a short description of the sampling design: "**Each sample was collected from a site that represent a unique combination of soil, climate and management. Specifically, between 9 and 30 soil cores were sampled in a 25 m x 25 m quadrat and split into two different depths, 0–0.1 m and 0.2–0.3 m, respectively, and then combined into one composite sample (approximately 1 kg of soil) for each depth.**"

P3 L 27-33: The soil maps used were a result of spatial modelling. The outcome of a model cannot be considered as data; please therefore change the word 'data' (L27 and L32) to 'values' or 'information'.

This is a good point. We have changed the word "data" to "values" or "information" throughout the manuscript, as suggested. We think all the continental soil maps currently available are a product of spatial modeling. However, these maps seem to be generally accepted and used as "data" rather than "information". Probably, these values are considered as data because of repeated model evaluation with real measurements. The European soil database and the SSURGO data are good examples.

P5 L8-10: The values given in these lines (sum of $NH_4^+$ and $NO_3^-$, $NH_4^+$ and $NO_3^-$) seem to refer to mean values (for the first of this set of values, it is indeed stated so). Would it not make more sense to give median values here, given that median values are what are shown on the corresponding graphs (figure 2)? The mean average value of a population that is not normally distributed is not particularly informative. Additionally, given that the $NH_4^+$ and $NO_3^-$ concentrations are not normally distributed, stating the standard deviation of these data is misleading, as the use of a SD value to convey information assumes the population is more or less normally distributed.

We agree. Please note that Figure 2 now shows box-whisker plots. We have reported the median values instead of the mean values, except for the mean comparisons. We also agree that the use of SD is not ideal for non normal data. Thus, we have reported the interquartile ranges, instead of SD values, to better describe skewed data. And we think we can avoid its misuse as the distribution of soil mineral N values is also shown in Figures 2, 4-6.

P5 L 19-13: Why carry out an analysis omitting the samples for which $NO_3^-$ concentration was below the detection rate? If a soil sample has a $NO_3^-$ concentration below the detection level, this does not equate with 'no data', but rather means that the $NO_3^-$ concentration is simply very low (as the authors indeed assume). Unless something specific is being tested, which I do not think is the case here, an analysis with these points removed is uninformative. This subject is re-visited in the discussion (P7 L11-12) but as the text is written, I still do not understand what information this analysis brings. The results of this extra analysis do however indicate that the agricultural soils have a bimodal distribution with respect to $NO_3^-$ concentration, i.e. many soils have very low concentration and many soils have a very high concentration (indicative of high addition rates of $NO_3^-$). If this additional analysis was carried out to illustrate this, the authors need to make this clear in the discussion, and indeed expand this point in the discussion.

We used all the data for our analysis and modelling – values below detection limit were coded as 0.5 mg N kg$^{-1}$. Our extra analysis do not indicate a bimodal distribution of nitrate concentrations across the soil in the production region. The $NO_3^-$ concentrations still follow a skewed distribution to the right, with or without these low values under the detection limit. We simply wanted to show a shift in the median (or mean) concentrations in soil mineral N budget, which is important for a soil N management perspective.

P7 L18-19: What is meant here by this sentence, particularly the term "agricultural soil"? Do the authors mean that not every site in the agricultural region is under agriculture? If so, please change accordingly and change the terminology to 'soil under agriculture'. If not, please explain the term 'agricultural soil'.

We apologize for the confusion. Now we have termed the two regions as "agricultural" and "non-agricultural" as we think this is more clear-cut. We hope that this change in the terminology makes regional comparisons by broad land use more comprehensible. We clearly define these two regions in the first sentence of the section 2.2: "**(2) the samples from the sites that originate from dryland and irrigated cropping, and from improved and native pastures used for animal grazing (hereafter referred to as the "agricultural" region), and (3) the samples from the sites that were conserved and in natural environments outside of the agricultural production zones (referred as the "non-agricultural" region)**".

P8 L2-3: This sentence is either incorrect (as I have understood it) or imprecisely written: I understand from this sentence that the $NH_4^+$ concentrations between the soils from the agricultural and ecological regions are different. According to figure 2 and text in the results section (P5 L8-9) however, $NH_4^+$ concentrations are similar. I suspect a more complex pattern is meant by the authors; this needs to be more clearly explained.

Thank you for picking that up, this was our mistake. The sentence has been corrected to "**As shown above, the amount of $NO_3^-$ or the sum of $NH_4^+$ and $NO_3^-$ …**".

P9 L 18-25: The text here is difficult to understand. I have a few suggestions that might help:

We sincerely appreciate the suggestions.

L 19: Replace "by each specific region" by "within each region" (if I have understood correctly).

We have followed the suggestion.

L20: Replace "on each region specific basis" with 'for each region' (if I have understood correctly).
The sentence L20-22 is very unclear. Is it referring to the higher prediction error for the high concentrations of $NO_3^-$ (in the ecological region in particular)? Rewrite.

We have followed the suggestion. The following sentence has been rewritten as: "**In contrast, the model performance of soil mineral N was substantially limited by high prediction error, particularly over a high range of concentrations at all sites and in the production region.**"

L23: "presence of small values" is too vague. Do the authors mean that the high frequency of samples with very small $NO_3^-$ concentrations is the cause of the limited overall model performance? Needs to be explained.

We have modified "presence of small values" to "**presence of samples with $NO_3^-$ concentrations under the detection limit**".

L24: Replace "most of model errors" with "much of the model error"

We have followed the suggestion.

L24-26: This sentence is too vague. This sentence relates to the presence of small values of what is meant be "the limited data set" exactly?

We agree - it is too vague. We have re-written the sentences as: "**As a result, much of the model error resulted from the lack of accuracy. In addition, the models may not capture all the processes and resulting variation as they were based on the limited data sets.**"

P10 L7-9: The authors here imply that they have identified a process in the results, the "potential to maintain or increase $NH_4^+$ concentrations". This process is a possible explanation of the results they have found, but is not in itself a result. Please change text accordingly.

Thank you. As suggested, we have changed the sentence as: "**Despite this limitation, our results suggest that soil organic matter and its C:N:P stoichiometry may contribute to the potential to maintain or increase $NH_4^+$ concentrations …**".

P10 L33-34: The term "management" should be avoided here, unless the authors specify what they mean by management, as a term separate from and in addition to 'land use'.

We have deleted "management" as we have not addressed any management effect on soil mineral N.

P 11 L 2: The term "human modification" should be removed; this term implies some sort of scale or land use intensity (e.g. nutrient input levels), but this has not explicitly been investigated in this study.

We have removed "as subjected to different levels of human modification".

Figure 1: Three main regions are shown here, whereas two are considered in the text. I recommend that the number of regions considered should be consistent. Alternatively, if the three ecological regions were considered distinct enough to warrant their separation on the map, why not use three regions in the analysis?

We agree that the shading of three regions in Figure 1 is initially confusing because we analyse only two regions. However, the reason that we shaded three regions in Figure 1 rather than only two is that we want it to be clear to readers that we acknowledge the fact that within the 'agricultural' region there are two regions that differ in the intensity of management. As stated previously, we would have definitely preferred to separate the data into three regions, however, the paucity and sparseness of the dataset did not allow that. Thus, our broad land use regions: "agricultural" and "non-agricultural". We made this definition and out assumptions clear in the text and in the figure caption. We hope this clarifies the concerns of the reviewer.

Please note that we have made a new Table 1.

Technical corrections

P1 L11: It is unclear what the 'other' ecological region refers to.

This has been corrected to the "non-agricultural" region.

P3 L17-22: The sentences from L17 to 22 need to be moved out of this section; I suggest to the discussion.

We understand that this reads like discussion. However, we wanted to justify our assumptions.

P5 L5-6: In the first sentence of the results, the mean $NH_4^+$ and $NO_3^-$ concentrations are stated, referring to figure 2. However, in figure 2, the median concentrations are given. Please correct text (or change figure w2) accordingly.

We have now reported the median values with our new Figure 2. In the previous figure, both mean and median values were reported.

P5 L25: Change "large" to "high"

We prefer "large" to 'high' as this refers to 'contents'.

P5 L26: "in that" needs to be inserted between the words "…..environments or" and "used mainly…..".

This comment makes no sense to us. Thus, we have not followed the suggestion. We will address the comment if the reviewer is willing to provide some clarification.

P6 L16-17: This sentence belongs in the discussion.

Not necessarily. We described that the model estimation was not accurate for high mineral N concentrations.

P7 L8: Remove "In our case".

We have followed the suggestion.

P7 L7: Change "which may suggest" to "which suggests"

We have followed the suggestion.

P7 L22-23: Change "…which was considerably low in the soil from the agricultural region compared to the ecological region" to "… which was considerably lower in the soil from the agricultural region compared to that in the ecological region".

We have followed the suggestion.

P10 L6: Change "difficulties to directly measure" to "difficulties in directly measuring"

We have followed the suggestion.

P10 L10: Change "also indication that' to 'also an indication that"

We have followed the suggestion.

P10 L11: Unclear. What depends mostly on soil mineralogy? Soil input stoichiometric ratios or final soil elemental ratios?

We have re-written the sentence to make it clear. "**There is also an indication that final soil elemental ratios are less affected by soil input stoichiometric ratios than previously expected and depends mostly on soil mineralogy …**"

P10 L28-29: Unclear. I think this sentence needs to be re-written as: "Therefore, the importance of soil elemental interactions in determining the variation of mineral N at different spatial scales across and within various cropping and ecological conditions needs to be estimated." I have possibly interpreted this sentence incorrectly. If this is the case, what is "it" (L28)?

Thank you. We have followed the suggestion. "**Therefore, the importance of soil elemental interactions in determining the variation of mineral N at different spatial scales across and within various production and ecological conditions needs to be estimated.**"

Figure 2: Please clarify, do the error bars shown represent the standard deviation of the data values i.e. population, or the standard error of the mean?

In the previous figure, the error bars represent the standard deviation of the data values. We made new box and whisker plots in Figure 2. Its caption has been changed to "**Figure 2. Mineral N contents (mg N kg$^{-1}$) and fractions of total N in soil. The bottom, middle and top of each box represents the 25th, 50th (median) and 75th percentiles, respectively. The points above the whiskers are extreme values. Means between main agricultural and non-agricultural regions of Australia are significantly different at P-value < 0.001 (\*\*\*), if indicated based on ANOVA on the log of the values.**"

Figure 4: Is 'standard error of the (estimated) mean' meant here?

In the caption, "the estimated mean" has been corrected to "**the estimated mineral N concentration**".

Additional technical corrections:

We have added the units of measured soil properties of the BASE data and values from the soil maps in the section 2.1. $NH_4^+$ and $NO_3^-$ were reported in mg N/kg. Organic C, total N, total P, and texture were reported in %. Bulk density and exchangeable CEC were reported in g/cm$^3$ and meq/100 g, respectively. Thus, we have corrected the term "concentration" to "content" for soil mineral N as determined by the mass of sample throughout the manuscript.

---

## Author Comment (AC4) · 6 Jul 2018

**Supplementary Table 1.** Definition of land use classes in the data set (469 sites), based on the Australian Land Use and Management Classification.

| Broad land use | Detailed land use | Definition | n |
|---|---|---|---|
| Production from dryland agriculture and plantations | Cropping | Land that is under cropping | 16 |
| | Environmental forest plantation | Area managed for environmental and indirect production uses | 12 |
| | Grazing modified pastures | Pasture, both annual and perennial, for grazing | 45 |
| | Native/exotic pasture mosaic | Pastures followed by extensive active modification or replacement of native vegetation | 3 |
| | No defined use | Land cleared of intact native vegetation where the proposed land use is not known | 1 |
| | Plantation forests | Land on which plantations of trees or shrubs | 1 |
| | Seasonal horticulture | Crop plants living for less than two years that are intensively cultivated | 3 |
| Production from irrigated agriculture and plantations | Irrigated cropping | Land that is under irrigated cropping | 3 |
| | Irrigated grapes | Irrigated grapes | 2 |
| | Irrigated sugar | Irrigated sugar | 5 |
| | Irrigated vine fruits | Irrigated vine fruits | 1 |
| Production from relatively natural environments | Grazing native vegetation | Land under native vegetation, used for grazing by domestic stock | 45 |
| | Production native forests | Commercial production from native forests | 23 |
| Conservation and natural environments | Biodiversity | Area managed for biodiversity | 1 |
| | Defence land - natural areas | Stock reserves under intermittent use or unused | 1 |
| | Habitat/species management area | Protected area managed mainly for conservation through management intervention | 10 |
| | Managed resource protection | Protected area managed mainly for the sustainable use of natural ecosystems | 3 |
| | National park | Protected area managed mainly for ecosystem conservation and recreation | 171 |
| | Natural feature protection | Protected area managed for conservation of specific natural features | 21 |
| | Other conserved area | Land under forms of other nature conservation protection | 18 |
| | Other minimal use | Area of land that are largely unused | 7 |
| | Rehabilitation | Land under rehabilitation that has been restored to a near natural state | 3 |
| | Residual native cover | Land under native cover, mainly unused or used for non-production or environmental purposes | 60 |
| | Stock route | Stock reserves under intermittent use or unused | 1 |
| | Strict nature reserves | Protected area managed mainly for science | 4 |
| | Surface water supply | Area managed as a catchment for water supply | 1 |
| | Traditional indigenous uses | Crown land managed primarily for traditional indigenous purposes | 6 |
| | Wilderness area | Protected area managed mainly for wilderness protection | 2 |

Source: ABARES 2016, The Australian Land Use and Management Classification Version 8, Australian Bureau of Agricultural and Resource Economics and Sciences, Canberra. CC BY 3.0.

[Figure]

Supplementary Figure 1. Importance of soil properties as the predictors of NH$_4^+$ contents (mg N kg$^{-1}$). The importance of the predictors is based on the usage of each variable in the Cubist model (black bars). None of soil properties is used to set the rule conditions.

[Figure]

Supplementary Figure 2. Importance of soil properties as the predictors of NO$_3^-$ contents (mg N kg$^{-1}$). The importance of the predictors is based on the usage of each variable in the rule conditions (grey bars) and in the Cubist model (black bars).

[Figure]

Supplementary Figure 2. Continued.

---

## Author Comment (AC5) · 26 Jul 2018

Topical Editor Decision: Revision (19 Jul 2018) by Axel Don

Comments to the Author:

Dear Dr Lee and Co-authors,

Two reviewers have evaluated the manuscript very carefully and in detail and suggested several points to improve the manuscript. Thank you for taking up most of these points in your reply and a revised version. Please not that a revised version is generally only due after the paper discussion period and the editorial board decision on the paper.

Two major aspects (raised also by the reviewers) need some further attention in order to make this paper consistent and sound:

We appreciate the comments from the topical editor. Please find our responses below. All changes have been marked in the revised manuscript.

1.) The classification into agriculture and non-agriculture requires better justification. The main management driver for mineral N in soils is fertilisation. Agricultural land that does to receive fertilisation, e.g. some grazed land, may therefore behave more like non-agricultural land than like agricultural land.

We fully agree that fertilizer application is an important management driver for soil $NH_4^+$ and $NO_3^-$, and this is the case especially for cropping systems and improved pasture in Western Australia, South Australia, Victoria and southern New South Wales.

As pointed out, some sites under production from relatively natural environments receive no external N input. Soil controls and mineral N dynamics in semi-natural pasture may be similar to those in the non-agricultural region, despite disturbance by grazing animals. There would be little inputs through excretion by animal (e.g., dung and urine) across the vast area of grazing pasture. Based on the BASE data, these sites are located in central Australia (see Figure 1) and characterized by low vegetation and soil N stocks in arid- and semi-arid climate. Nevertheless, the majority of the N that animals ingest from herbage is returned to the pasture via excreta (e.g., at least 75% by cattle, see Whitehead 1995), which is a key component that influences the distribution of soil mineral N. Production from relatively natural environments includes forested agricultural land as well, where considerably larger biomass input occurs during commercial production from forests. Thus, these sites may behave more like intensive agricultural land as opposed to semi-natural pasture. Please note the term "forested" mean "trees" or "sparse trees" with a crown cover of > 20% according to the recent Land use of Australia.

Due to the scarcity of data and the difficulty in grouping land use classes, we were not able to further detail broad land uses by main management driver, such as fertilization. In this study, we have used the classification of broad land uses according to the Australian Land Use and Management Classification, which seem to be the most objective.

This is why we decided to analyze and simulate the data by the two regions, "agricultural" and "non-agricultural". Again, this decision was made by full consideration of data size for each of the broad and detailed land uses (see Table 1 and Table S1).

To further justify our classification, we have included the following statement following our definition of the regions: "**A total of 469 sites were retained for 10 data analyses (Fig. 1), and we performed the analyses on (1) all samples, (2) the samples from the sites that originate mainly from dryland and irrigated cropping, and from improved and native pastures used for animal grazing (hereafter referred to as the "agricultural" region), and (3) the samples from the protected sites and those in natural environments outside of the agricultural production zones (referred as the "non-agricultural" region). The agricultural region (160 sites) covers the main grain-cropping zones of Australia, which differs by climate and soil regimes, and farming practices. Sites used for**

**agricultural production from relatively natural environment did not receive external N fertilization but some N excretion by grazing animals or biomass inputs from commercial production.**"

2.) I agree with reviewer 2 that climate and geology are important drivers for continental scale variability in soil properties. If this study ignores these large scale drivers (primary drivers) but focusses on "proximal drivers" such as TOC and pH, the manuscript has to be revised to clarify and explain this to the reader. This refers to the whole manuscript and particular on the title and the objectives.

Climate and geological attributes are potentially important for continental scale distribution of soil mineral N. And we did not ignore these factors. In fact, they were considered to predict the soil properties from the soil maps in our spatial simulations.

In the section 2.1, we stated "**The approaches used to produce the maps were described in detail in Viscarra Rossel et al. (2015). When the soil maps were produced at the fine spatial scale, auxiliary environmental data were already considered in the spatial modeling. These variables represent proxies for the main environmental factors of soil formation, which were related to parent material, climate, biota and vegetation, and terrain and landscape position**."

More importantly, we intended to assess soil controls over soil mineral N concentrations without confounding effects of climate or other variables in Cubist modeling. In addition, the coarse resolution of climate data (usually 25 km after downscaling) does not match with the fine resolution of soil measurements (25 m).

As suggested, we have changed the title as: "**Continental soil drivers of ammonium and nitrate in Australia**".

Please note that our objectives are set out very clearly - we have focused on soil controls only. Throughout the manuscript, our results and discussion are limited to the potential controls of 10 soil properties obtained from the BASE data and our recent soil maps. We went over the whole manuscript again to make sure that we are following this comment.

Please also provide more details on the actual analysis and extraction method of nitrate and ammonia that was used.

After soil drying and sieving (< 2 mm), the concentrations of mineral N were determined colorometrically after extraction with 1 M potassium chloride. We have added the sentence: "**The contents of mineral N were determined colorimetrically on dried soil samples < 2 mm after extraction with 1 M KCl.**"

We are looking forward to a revised version of your manuscript.

Yours sincerely
Axel Don

Additional reference

Whitehead DC (1995) 'Grassland nitrogen.' (CAB International: Wallingford, UK)

---

## Author Response (AR2)

Topical Editor Decision: Publish subject to minor revisions (review by editor) (27 Aug 2018) by Axel Don
Comments to the Author:
Dear Dr Lee and Co-athors,

Thank you for uploading the revised version of your paper. It improved and is becoming almost ready to be published in SOIL.

Please clarify one issue on the non-agricultural land:
On page 4, l. 13 please add if there was (light) grazing by sheep or cattle also possible at sampling sites classified as "non-agricultural land". This might be important for the reader to follow the classification scheme.

Best regards,
Axel Don

We sincerely appreciate the comment from the topical editor. As suggested, we have added the following sentence to make clear that there was no commercial grazing at the non-agricultural sites. Specifically now on page 4 line 16: "There was no livestock grazing in the non-agricultural region."